# Validity and Reliability of the Caregiver Strain Index Scale in Women during the Puerperium in Spain

**DOI:** 10.3390/ijerph18073602

**Published:** 2021-03-30

**Authors:** David Feligreras-Alcalá, María del Pilar Cazalilla-López, Rafael del-Pino-Casado, Antonio Frías-Osuna

**Affiliations:** Department of Nursing, Faculty of Health Sciences, University of Jaén, 23071 Jaén, Spain; dfa00001@red.ujaen.es (D.F.-A.); mpcl0008@red.ujaen.es (M.d.P.C.-L.); afrias@ujaen.es (A.F.-O.)

**Keywords:** subjective burden, caregiver strain index, puerperium, postpartum depression, validity, reliability

## Abstract

Background: The objective of this study is to determine the validity and reliability of the Caregiver Strain Index (CSI) for women during the postpartum period. Methods: This is a validation study of a measurement instrument. This study includes 212 women over the age of 19 who gave birth from March to September 2019 in Maternal and Child Hospital of Jaén (Spain). The items of the CSI were adapted for newborn care. Content validity was measured by five experts, calculating the index of agreement (Aiken’s V). Criterion validity was assessed by correlations with scores of other tools that measure constructs related to burden (Edinburgh Postpartum Depression Scale, State-Trait Anxiety Questionnaire, SOC-13 and Duke-UNC-11). Construct validity was determined by the known-groups method. Internal consistency was measured using Cronbach’s Alpha, and stability was analysed using the intraclass correlation coefficient (ICC). Results: Regarding content validity, an Aiken’s V of 1.00 (*p* = 0.032) was obtained. Regarding criterion validity, the correlation analyses showed statistically significant coefficients between the scores of the questionnaire and those of the sense of coherence (r = −0.447, *p* < 0.001), depressive symptoms (r = 0.429, *p* < 0.001), social support (rho = −0.379, *p* < 0.001) and anxiety symptoms (r = 0.532, *p* < 0.001). The known-groups method showed statistically significant differences in the mean of subjective burden between the groups (depressive symptoms, anxiety symptoms, sense of coherence and social support). The total scale obtained a Cronbach’s alpha value of 0.710. The ICC was 0.979. Conclusions: The adapted CSI is a valid and reliable screening tool for the subjective burden in women during the puerperium. The adapted CSI can play an important role as a guide to detect the subjective burden in women during the puerperium.

## 1. Introduction

The process of pregnancy, childbirth and puerperium constitutes an important event in a woman’s life, where she faces significant changes in her cognitive, behavioural and social spheres [1]. Sometimes, this process can be perceived as stressful [2,3], which can lead to an increase in the subjective burden on newborn care [4,5]. The subjective burden has been defined based on the model of the General Theory of Stress. In this model, the caregiver must face certain stressors, and the caregiver’s response will be conditioned by psychological processes that include emotional impact, the perception of social support and coping strategies [6]. On the other hand, according to the Lazarus and Folkman Stress Transactional Model, the consequences of stress are measured according to the way in which caregivers perceive, evaluate and manage the care process [7].

Thus, subjective burden is considered a state characterised by stress, fatigue and difficulties in adapting to the role of caregiver, caused by a negative evaluation of the caregiver’s situation that threatens their physical, psychological and emotional health [8]. The term subjective burden refers to the assessment by the caregiver that the care situation surpasses the skills that the caregiver possesses to deal with the situation adequately [9]. In this sense, it can also refer to the perception that women have of feeling bewildered, burdened, trapped, resentful and excluded [9]. The objective burden is considered to reflect the daily and practical aspects of the provision of care that capture quantitative dimensions of the caregiver function, such as the level of care needs and the hours of care provision [10].

The presence of caregiver burden can be associated with the appearance of disorders such as depressive symptoms [11] and anxiety [12]. In this context, we could consider the puerperium as a stage in which the subjective burden is significant. Some authors affirm that women with high levels of care burden could have a greater tendency to develop health problems during the puerperium. The investigation of factors associated with the presence of negative consequences related to the subjective burden during the postpartum period, takes on special importance in the early detection and prevention of these factors, such as anxiety [13] or depressive symptoms [14].

Several studies have been carried out to explore the possible association of subjective burden and depressive symptoms in caregivers of the elderly [15,16,17]. In relation to the presence of postpartum depression (PPD), there are studies that suggest that a higher level of subjective burden regarding newborn care could be associated with a higher level of stress and an increase in depressive symptoms [18,19]. Similarly, Leung et al. [14] show the importance of stress with newborn care as an important predictor of the appearance of PPD. Postpartum depression has a prevalence of 12–13% at 6 weeks of postpartum in industrialised countries [20,21]; in Spain, the studies carried out place it between 10% and 23% [22,23,24,25].

In this sense, it would be especially useful to have a measurement instrument adapted to women in the puerperium, which allows a more specific assessment of the construct of the subjective burden of care in this population. The Caregiver Strain Index (CSI) is an instrument designed to measure the subjective burden of caregivers of dependent family members. It is a simple scale, which requires little time to complete, and offers the possibility of an empathic approach towards the caregiver [26]. Robinson validated this instrument in 1983 in the United States (U.S.) in caregivers of patients, after hospital discharge, with atherosclerotic heart disease or with implantation of a hip prosthesis [27].

The CSI scale has been widely used and adapted to diverse populations, cultural contexts and languages [28,29,30]. In the Spanish population, this scale has been validated with caregivers of patients with chronic, oncological and acute pathologies that required home care [31] and in caregivers of patients diagnosed with dementia [26], with an acceptable internal consistency (Cronbach’s alpha of 0.80). This scale is made up of 13 items with a dichotomous response (yes/no) in the context of a semistructured interview whose score ranges from 0 to 13 points [31]. However, no validation study has been found to prove its use in women during the puerperium.

Thus, the adaptation of this CSI scale would have enormous clinical applicability. On the one hand, it would allow the detection of subjective burden in these types of caregivers and early identification of women with high levels of subjective burden in care and therefore with a tendency towards postpartum depression or anxiety, among other associated complications. On the other hand, it would allow improving the treatment of these health problems with a more specific and individualised approach. Thus, this scale would become a useful tool for detecting the subjective burden in care in puerperal women. The aim of this study is to determine the validity and reliability of the CSI in women during the puerperium.

## 2. Materials and Methods

### 2.1. Questionnaire Adaptation

To carry out this work, the items of the CSI scale (Spanish version validated by López Alonso et al. [31]) were adapted for newborn care, after consensus of a working group formed by the authors of the present study.

### 2.2. Content Validation by Experts

Content validity was measured by five experts from the Department of Nursing of the University of Jaén, with extensive academic experience in the study area and other complementary areas. The index of agreement that should exist to determine this validity was established using the Validity Coefficient V of Aiken [32,33].

### 2.3. Pilot Test

A pilot test was conducted to determine the comprehension and applicability of the scale through semistructured interviews with 40 women during the puerperium, during which they were asked if they adequately understood the items of the scale. Subsequently, an analysis and interpretation of the responses obtained was carried out, identifying possible aspects that were not well understood.

### 2.4. Clinical Validation 

Clinical validation was carried out through a descriptive cross-sectional study in women during the puerperium in the province of Jaén, Spain, using the data collected by Feligreras et al. [34]. Two-hundred-and-twelve women who gave birth at the Maternal-Infant Hospital of Jaén were recruited by random sampling between March and September 2019.

The sample size analysed allows us to affirm, for a 13-item scale, that the calculated Cronbach’s alpha is significantly higher than the value of 0.7, with a statistical power of 80% and a significance level of 5%, taking the value of the null hypothesis at 0.6 (calculations performed with PASS 11). The aforementioned sample size also allows us to detect differences of at least 1.1 points out of 13 (8.5%) with a power of 80% and a significance level of 5%, taking as a reference a standard deviation of 2.76 (calculations made with EpiDat 4.2).

The exclusion criteria were age equal to or less than 19 years, previous and/or current personal history of psychiatric pathology, serious illness or death of the newborn, not understanding the Spanish language, not accepting participation in the study or not signing the informed consent.

For the characterisation of the sample, the following variables were collected: age, marital status, educational level, employment situation, family income, pregnancy search, number of pregnancies, type of delivery, sex of the newborn and family history of psychiatric pathology.

#### 2.4.1. Criterion Validity

Criterion validity was assessed by correlations with the scores of other instruments that measure constructs related to subjective burden. These measurement instruments were the Edinburgh Postpartum Depression Scale (EPDS) for the depression construct [35], the state-trait anxiety scale STAI for the anxiety construct [36], the Antonovsky SOC scale for the construct of sense of coherence [37] and the Duke-UNC-11 scale for the construct of social support [38].

Depressive symptoms were measured using the EPDS [35]. This scale is used to detect depressive states in the postpartum period. It is a self-administered scale of 10 items, with four possible response alternatives, scored from 0 to 3, depending on the severity of symptoms. Scores range from 0 to 30 points (proportional to the level of depressive symptoms). A cut-off point, equal to or greater than 10 points, is considered adequate to detect depressive symptoms in this period, with a sensitivity of 79%, a specificity of 95% and a positive predictive value of 63% [25]. Its use is recommended in the first 6 weeks of postpartum to ensure correct screening for depressive symptoms in puerperium, according to the Ministerio de Sanidad, Consumo y Bienestar Social of Spain [39]. It is validated and widely applied in Spain [25].

Anxiety was measured by STAI state-trait’s anxiety questionnaire [36], a self-administered instrument that measures two independent concepts of anxiety: On the one hand, anxiety as a state is referred to as a transient emotional condition; on the other hand, anxiety as a trait is described as a relatively stable anxious propensity. In the development of this research, anxiety status has been evaluated as a specific measure of anxiety during this study period. This subscale consists of 20 items, with 4-point, Likert responses (proportional to intensity of anxiety). Total subscale scores range from 0 to 60 points, with the 75th percentile recommended as a cut-off point in adult women [36]. This questionnaire is validated in the Spanish population, presenting a Cronbach’s alpha coefficient of 0.94 [40].

The sense of coherence was collected using the SOC-13 Sense of Coherence scale [37]. This scale assesses three dimensions of sense of coherence: compressibility, manageability and significance, which are closely related. It consists of 13 items that are answered on a Likert scale with seven scores, ranging from 1 (always) to 7 (never), where a higher score indicates a greater sense of coherence. This scale is validated in the Spanish population, presenting a Cronbach’s alpha of 0.80 [41]. 

Perceived social support was measured by the Duke-UNC-11 questionnaire [38], a self-administered instrument that measures social support both in its affective dimension (i.e., referring to expressions of love, appreciation, sympathy or belonging) and its confidential dimension (i.e., that through which people can receive information, advice or guidance). It consists of 11 items along with a 5-point, Likert response scale, ranging from 1 (Much less than I want) to 5 (As much as I want). Scoring for the total questionnaire ranges from 11 to 55 points (directly proportional to the level of perceived social support). This questionnaire has been validated in the Spanish population with adequate psychometric properties (e.g., Cronbach’s alpha coefficient of 0.93) [42].

These measurement instruments were chosen because there is sufficient scientific evidence that relates the subjective burden with a higher incidence of depression in postpartum women [19], with higher levels of anxiety in caregivers of patients at the hospital level [43], with a lower sense of coherence in caregivers of dependent elderly relatives [11] and with less social support in caregivers of adults with schizophrenia [44].

We used the Pearson bivariate correlations between the scores of the adapted CSI and the scores of the questionnaires of the variables that fulfilled the assumptions of normality (depressive symptoms, anxiety and sense of coherence), and bivariate Spearman correlations in social support; this was a similar analysis to that carried out in the validation of the CSI in our country [31]. 

#### 2.4.2. Construct Validity

To evaluate the construct validity, the known-groups method was performed. Difference of means tests were carried out using the Student’s t-test and the magnitude of the effect was measured using Cohen’s d. The hypotheses tested were, according to the scientific literature on subjective burden, (1) the greater the presence of depressive symptoms, the higher the subjective burden level in women in the puerperium [19]; (2) the greater the presence of anxiety symptoms, the higher the level of subjective burden in women in the puerperium [43]; (3) the lower the sense of coherence, the higher the level of subjective burden [11]; and (4) the less social support, the higher the level of subjective burden in women in the puerperium [44]. The cut-off points proposed by each author in the scales used for depressive symptoms, anxiety symptoms and social support were used, whereas we used the median in the case of the sense of coherence.

#### 2.4.3. Reliability

##### Internal Consistency

To determine internal consistency, Cronbach’s alpha was analysed. For the assessment of this coefficient, the recommendations of George and Mallery [45] were used, who show that an acceptable internal consistency has values greater than 0.70.

##### Stability (Test–Retest)

Stability was measured with the intraclass correlation coefficient (ICC). To do this, the scale was administered again after one week to a sample of 20 women. Following the Fleiss recommendations [46], values above 0.75 represent excellent reliability.

To carry out the analyses, the level of statistical significance was set at 0.05. The relationships between the proposed variables were examined with the help of the SPSS v. 22.0 (IBM International Business Machines Corporation, Armonk, NY, USA) program.

## 3. Results

### 3.1. Definitive Questionnaire and Pilot Test

In adapting the questionnaire, the examples that the authors provided in the original version for chronic, oncological and acute patients requiring home care were replaced with examples related to newborn care. In addition, item number 10 has been adapted to a newborn care situation. The final questionnaire (the CSI adapted to newborn care) is showed in Appendix A, and its translation into English is presented in Appendix B.

Regarding content validity, the five experts obtained a Validity Coefficient V of Aiken of 1.00 (*p* = 0.032) for the total of the items of the questionnaire [32,33]. Therefore, according to the recommendations of Aiken, we can show the maximum agreement among the experts, avoiding random coincidences. There were no changes derived from the performance of the content validity analysis in the questionnaire.

After conducting the pilot test, an adequate understanding of the item statements was verified.

### 3.2. Clinical Validation

Descriptive data of the sample are shown in Table 1. The mean age of the participating women was 32 years, with a minimum age of 19 and a maximum age of 47 years. Most of the women were married (78.3%) with an active employment situation, self-employed or employed (69.4%). Forty-seven percent of the participants had university studies, and 90.1% of the women reported that the pregnancy had been wanted.

The minimum CSI score obtained was 0 and the maximum score was 13, the mean score being 5.23 (SD: 2.76). The correlations between the different items and the overall result were higher than 0.2 in all cases [47], and the ICC for the agreement between items was 0.71 [48].

### 3.3. Criterion Validity

The correlation analyses showed statistically significant coefficients between the scores of the adapted CSI questionnaire and those of sense of coherence (r = −0.447, *p* < 0.001), depressive symptoms (r = 0.429, *p* < 0.001), social support (rho = −0.379, *p* < 0.001) and anxiety symptoms (r = 0.532, *p* < 0.001). 

### 3.4. Construct Validity

We proceeded to perform the known-groups method. There were statistically significant differences in the mean of subjective burden between the groups analysed: (1) presence or not of depressive symptoms (6.68 vs. 4.69; d: 0.77; 95% CI: 0.45, 1.08), (2) presence or absence of anxiety symptoms (7.08 vs. 4.51; d: 1.02; 95% CI: 0.71, 1.34), (3) presence or not of sense of coherence (4.28 vs. 6.21; d: −0.76; 95% CI: −1.04, −0.48) and (4) presence or not of social support (4.88 vs. 7.00; d: −0.81; 95% CI: −1.18, −0.43) (Table 2).

### 3.5. Reliability

In relation to internal consistency, Cronbach’s alpha was determined for the total scale, obtaining a value of 0.710 (acceptable internal consistency). Regarding the stability of the test, the Intraclass Correlation Coefficient (ICC) was measured, reaching a value of 0.979 (95% CI = 0.949–0.992) (excellent stability).

## 4. Discussion

In this study, we presented the adaptation and validation of the CSI in order to measure the subjective burden in women during the puerperium. The CSI adapted to newborn care was quick and easy to administer in the population studied. The study participants adequately understood each of the items both in the pilot test and in the final application of this measurement instrument. The adapted scale obtains adequate psychometric properties at the level of criterion validity and construct validity, internal consistency and stability, which shows an adequate validity and reliability of the measurement instrument.

Criterion validity was assessed using bivariate correlations with scores from other instruments that measured constructs related to subjective burden. López-Alonso et al. [29] in the CSI validation in the Spanish population carried out a similar analysis, correlating the subjective burden with other similar areas included in the original validation of the scale.

Regarding the construct validity, we confirmed all the hypotheses tested in the known-groups method, which reinforces the construct validity of the instrument in measuring subjective burden in women during the puerperium.

Regarding internal consistency, we can find some coincidences of our study with other studies on CSI in other types of caregiver populations. Thus, Ugur et al. [49] carried out an adaptation of this scale in the Turkish population, finding a Cronbach’s alpha value of 0.77. Ramasamy et al. [50] in Malaysia obtained an internal consistency of 0.75. The Chinese version [51] showed high reliability with a Cronbach’s alpha of 0.91, the same result as that obtained in the Portuguese population [52]. In our study, we obtained an internal consistency of 0.710. Therefore, our results in this regard are similar to those available in the literature.

Regarding the measurement of stability, the adapted CSI was administered again one week after the first measurement in the puerperium; the time elapsed between both applications should not be too long (variation of the phenomenon studied) nor too short (learning effect) [53]. The ICC for test–retest reliability was 0.979, higher than the 0.88 obtained by Thornton and Travis in the modified version of the CSI [54]. Both in the original validation of the CSI [27] and in the validation of the CSI in our country [31], test–retest reliability was not calculated. Following Fleiss [46], values above 0.75 show excellent reliability. These data confirm the stability of the CSI adapted to newborn care.

To date, in the scientific literature there was no validation of the CSI scale in women during the puerperium. In this way, with the adaptation and validation of this scale, we obtain a measurement instrument that has enormous clinical applicability. This scale allows us to measure in a more specific way the construct of the subjective burden of care in women during the puerperium. In this sense, the adapted CSI in the puerperal population offers us the possibility of early diagnosis of women with high levels of subjective burden and, therefore, with a possible higher risk of developing PPD or anxiety (among other associated complications). Therefore, this scale constitutes a useful tool for detecting the subjective burden of newborn care in puerperal women.

The study has the limitation that the entire sample obtained for the study comes from a single hospital centre. Another limitation is that the study does not fully comprise some perinatal factors and circumstances associated with delivery, which might affect the evaluation of the criterion validity.

## 5. Conclusions

According to the findings obtained in this research, the CSI adapted to newborn care is a valid and reliable screening tool for the subjective burden in women during the puerperium. This study can play an important role as a guide to detect the subjective burden in women during the puerperium, in order to plan preventive and health promotion actions in those women with a risk profile of subjective burden in newborn care.

## Figures and Tables

**Table 1 ijerph-18-03602-t001:** Description of the studied sample.

Variables		*n* (%)	M (SD)	CI 95%
Age			32.670 (4.58)	32.06–33.26
Marital Status	Single	8 (3.8)		1.40–6.60
	Married	166 (78.3)		72.60–83.50
	With couple	35 (16.5)		11.80–21.70
	Separated or divorced	3 (1.4)		0.00–3.30
Education level	Primary	18 (8.5)		4.70–12.30
	Secondary	21 (9.9)		6.10–14.20
	High School	12 (5.7)		2.40–9.00
	FP Middle degree	36 (17.0)		12.30–22.20
	FP Higher degree	24 (11.3)		7.10–15.60
	University	101 (47.6)		41.00–54.20
Employment situation	Student	3 (1.40)		0.00–3.30
	Active or own account	39 (18.4)		13.70–23.60
	Asset	108 (50.9)		43.90–57.10
	Unemployed	47 (22.2)		16.50–27.80
	Domestic work	15 (7.1)		4.20–10.80
Family income	<from 500€	1 (0.5)		0.00–1.40
	From 500 to <1000€	27 (12.7)		8.50–17.00
	From 1000 to <1500€	60 (28.3)		22.60–34.40
	From 1500 to <2000€	54 (25.5)		19.80–31.60
	From 2000 to <2500€	29 (13.7)		9.00–17.90
	From 2500 to <3000€	28 (13.2)		9.00–17.90
	From 3000 to <5000€	10 (4.7)		1.90–8.00
	>from 5000€	3 (1.4)		0.0–3.30
Pregnancy wanted	Yes	191 (90.1)		85.80–93.90
	No	21 (9.9)		6.10–14.20
No. of pregnancies			1.835 (0.986)	1.71–1.97
Type of delivery	Eutocic	141 (66.5)		60.80–73.10
	Instrumental	32 (15.1)		10.40–20.30
	Caesarean section	39 (18.4)		13.20–23.60
Sex of newborn	Male	112 (52.8)		46.20–59.90
	Female	100 (47.2)		40.10–53.80
Family history of psychiatric pathology	Yes	13 (6.1)		3.30–9.40
	No	199 (93.6)		90.60–96.70

Source: self-made.

**Table 2 ijerph-18-03602-t002:** Differences in means of subjective burden in the subgroups of women with and without depressive symptoms, anxiety, sense of coherence and social support.

Variable	Mean (SD)	Significance	Cohen’s d	95% CI
Depressive symptoms	Yes: 6.68 (2.72)	*p* < 0.0001	0.77	(0.45, 1.08)
	No: 4.69 (2.58)			
Anxiety	Yes: 7.08 (2.62)	*p* < 0.0001	1.02	(0.71, 1.34)
	No: 4.51 (2.47)			
Sense of coherence	Yes: 4.28 (2.56)	*p* < 0.0001	−0.76	(−1.04, −0.48)
	No: 6.21 (2.62)			
Social support	Yes: 4.88 (2.67)	*p* < 0.0001	−0.81	(−1.18, −0.43)
	No: 7.00 (2.52)			

Source: self-made.

## Data Availability

The data presented in this study are available on request from the corresponding author. The data are not publicly available due to their containing information that could compromise the privacy of research participants.

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
