# Peer review of "Validity and Reliability of the Caregiver Strain Index Scale in Women during the Puerperium in Spain"

_ijerph, 2021, doi:10.3390/ijerph18073602_

Round 1
Reviewer 1 Report
I thank the editors for the opportunity to collaborate as a reviewer in the International Journal of Environmental Research and Public Health. I would also like to congratulate the authors of the manuscript "Validity and Reliability of the Caregiver Strain Index Scale in Women During the Puerperium in Spain", for the effort made in their study.
I regret to say that due to certain important limitations it is not possible to accept the manuscript for publication:
- Lines 61 and 62. Reference is made to research with the elderly. It is out of context. It should be deleted.
- The Introduction should include data on the prevalence of postpartum depression in Spanish women.
- Insufficient information is provided on the adaptation of the items for newborn care. The working group should be made up of experts who were not part of the present study.
- The section on the pilot test does not include information on the aspects that were not well understood.
- In the inclusion criteria, they did not take into account the presence of anaemia and its impact on stress and fatigue.
- The coherence scale is validated in Spanish, but in elderly people over 70 years of age, which indicates the inadequacy of its use in the population of the present study.
- Line 172. "clinimetric properties" should be changed to "psychometric properties".
- Line 177. The rationale for the use of references to schizophrenic patients is not understood.
- For the use of parametric tests, it is necessary to perform the normality test and the homoscedasticity test. This information is not provided.
- At least an exploratory factor analysis or a confirmatory factor analysis should have been carried out.
- Cronbach's alpha of 0.710 is acceptable for questionnaires for research purposes, but is not sufficient for diagnosis, as indicated by Nunnally in the literature. A minimum of 0.9 is needed. The authors intend to detect and plan preventive actions, for this Cronbach's alpha is not sufficient.
- In the Discussion only the information in the Results is repeated. The Discussion section should be modified.
- They only point out a limitation in the study, without taking into account other aspects that may have influenced the results and that have not been consulted in the study (such as: taking medication, having family support, etc.).
- There are few updated references, it is recommended that at least 40% are from the last 4 years, to ensure that the information in the manuscript is up to date.
Reviewer 2 Report
Please note that I was not able to copy/paste. Therefore, I uploaded my review below.

Reviewer 3 Report
The article has a good scientific level. Summary, material and method, and discussion are correct. I do not find scientific weaknesses. In the Validity criteria perhaps the Cronbach values should be specified. Otherwise, the article can be published in its present form.
RJ
Reviewer 4 Report
Dear authors, congratulations for your important study.
Your manuscript is very well organized and with a very good scientific soundness.
Just suggest in the conclusions, make a more objective reference to the contributes of your study to:
Clinical Care;
Research;
Society.
The overall manuscript is very good. Congratulations again.
Round 2
Reviewer 1 Report
The authors have made the suggested changes. The manuscript can be published in its present form.
Author Response
Thank you very much for your comment.